# Quality of Life After Coronary Artery Bypass Surgery: A Systematic Review and Meta-Analysis

**DOI:** 10.3390/ijerph17228439

**Published:** 2020-11-14

**Authors:** Jacqueline Schmidt-RioValle, Moath Abu Ejheisheh, María José Membrive-Jiménez, Nora Suleiman-Martos, Luis Albendín-García, María Correa-Rodríguez, José Luis Gómez-Urquiza

**Affiliations:** 1Faculty of Health Sciences, University of Granada, Avenida de la Ilustración N. 60, 18016 Granada, Spain; jschmidt@ugr.es (J.S.-R.); moad.ibrahem@hotmail.com (M.A.E.); norasm@ugr.es (N.S.-M.); lualbgar1979@ugr.es (L.A.-G.); macoro@ugr.es (M.C.-R.); jlgurquiza@ugr.es (J.L.G.-U.); 2Institute of Health Management, University Hospital of Ceuta, C/Colmenar, s/n, 51003 Ceuta, Spain

**Keywords:** coronary artery bypass graft, meta-analysis, prevalence, surgery, systematic review, quality of life, follow-up study

## Abstract

Coronary heart disease is a public health problem and is one of the leading causes of loss of quality of life, disability, and death worldwide. The main procedure these patients undergo is cardiac catheterisation, which helps improve their quality of life, symptoms of myocardial ischemia, and ventricular function, thus helping increase the survival rate of sufferers. It can also, however, lead to physical consequences, including kidney failure, acute myocardial infarction, and stroke. The objective of this study was to analyse how coronary artery bypass grafting (CABG) influences quality of life. A systematic review and meta-analysis were conducted using the CINAHL, PubMed, Scopus, and Cuiden databases in June 2020. A total of 7537 subjects were included, 16 in the systematic review and 3 in the meta-analysis. The studies analysing quality of life using the SF questionnaire showed improvements in the quality of physical and mental appearance, and those using the NHP questionnaire showed score improvements and, in some cases, differences in quality of life between women and men. This operation seems to be a good choice for improving the quality of life of people with coronary pathologies, once the possible existing risks have been assessed.

## 1. Introduction

Coronary heart disease is a public health problem and is one of the leading causes of loss of quality of life, disability, and death worldwide. The main procedure these patients undergo is cardiac catheterisation, both for diagnostic and therapeutic purposes [1]. Cardiac catheterisation helps improve quality of life, symptoms of myocardial ischaemia, and ventricular function, thus helping increase the survival rate of sufferers. It can also, however, lead to physical consequences, including kidney failure, acute myocardial infarction, and stroke [2]. It also has psychological consequences, such as stress, anxiety, fear, and depression [3]. Another widely used therapeutic option for the treatment of coronary disease worldwide is coronary artery bypass grafting (CABG). The development of this technique in recent decades has led to an improvement in both postoperative and long-term outcomes [4].

People with coronary heart disease are more likely to suffer heart problems and other pathologies [5] as, in addition to being one of the main causes of death worldwide, it also favours the development of comorbidities. The treatments described above are designed to improve myocardial perfusion, to improve the symptoms of and reduce the incidence of heart attacks and death [5].

This improvement in cardiac activity is also reflected in the daily life of an individual with heart disease. In fact, improved quality of life is one of the most sought aspects. Evaluating quality of life allows us to ascertain a subjective assessment of an individual’s health, as well as the impacts that the disease and its treatment have on that person’s daily life [6].

Quality of life is a concept that encompasses the physical, emotional, and social dimensions, and it varies with time and the individual’s perception [7,8]. There are currently a multitude of questionnaires available to measure patient quality of life. The generic validated Short Form 36 (SF36) questionnaire has been used on patients undergoing heart surgery. It consists of 35 items, distributed across eight domains and is divided into two main groups: physical and psychological components [9]. 

Another questionnaire reported in the literature for measuring quality of life in patients with chronic diseases and disabling symptoms is the two-part Nottingham Health Profile (NHP) [10]. The first part contains 38 items, divided into six dimensions: physical mobility, pain, sleep, energy, social isolation, and emotional reactions. Patients answer yes or no to the questions according to whether they have suffered the problems. The second part comprises seven aspects affected by the patients’ health status: capacity to work, ability to do housework, social life, family relationships, sex life, hobbies, and holidays. The score for each section ranges from 10 to 100, the higher the score the greater the problem presented by the patient and the lower their quality of life. 

A CABG involves risk, so this study was designed to elucidate the benefits this operation provides people who undergo the surgery, for example, benefits related to their quality of life. Thus, the objective of this study was to analyse how coronary artery bypass graft (CABG) influences quality of life.

## 2. Materials and Methods

A systematic review including a meta-analysis was conducted, following the recommendations of the Preferred Reporting Items for Systematic Reviews and Meta-Analysis (PRISMA) statement [11]. 

### 2.1. Search Strategy

The search was carried out using the CINAHL, PubMed, Scopus, and Cuiden databases, between March and June 2020. The search equation, based on MeSH terms, was (“Quality of Life” OR “Health-Related Quality of Life” OR “Life Quality”) AND (“Aortocoronary Bypass” OR “Bypass Surgery, Coronary Artery” OR “Bypass, Coronary Artery” OR “Coronary Artery Bypass Grafting” OR “Coronary Artery Bypass Surgery”) AND “Postoperative Period”. No restrictions were placed on the publication date to minimize publication bias.

### 2.2. Inclusion and Exclusion Criteria

The inclusion criteria were quantitative studies, with subjects undergoing both elective and emergency coronary artery bypass graft surgery, and studies using validated scales, written in English, Spanish, or Portuguese that included post-surgery follow-up. The exclusion criteria used were studies that included subjects undergoing other types of cardiological operations, such as valve replacement; studies that did not use a validated questionnaire to measure quality of life or which exclusively measured psychological variables, such as anxiety or depression; studies that included paediatric, psychiatrically challenged, and intubated/sedated patients or those with language difficulties.

### 2.3. Selection Process and Result Codification

Two team members independently conducted the search, selection, and analysis of the studies found. In the event of a disagreement, a third researcher from the group intervened. The selection was based on a reading of the title and abstract, then the full text, and finally a reverse search in the selected studies. For the meta-analysis, we selected studies that used the same measuring instrument and provided the data necessary for its execution. More specifically, these studies used the second part of the NHP questionnaire, as it provides viable data for meta-analytical estimation. The variables studied were (a) authors; (b) year and country of publication; (c) type of surgery (emergency or elective); (d) characteristics of the sample, such as: number of subjects included, sex, and follow-up over time; (e) instrument for measuring quality of life; (f) scores on the quality of life scale.

### 2.4. Critical Reading and Level of Evidence

The studies included in the research were reviewed critically for bias analysis, using the Strengthening the Reporting of Observational Studies in Epidemiology (STROBE) checklist [12]. The selected studies were assigned a methodological quality grade according to the levels of evidence and degrees of recommendation proposed by the Working Group on Levels of Evidence of the Oxford Centre for Evidence-Based Medicine (OCEBM) [13].

### 2.5. Statistical Analysis

Random effects meta-analyses were performed using the StatsDirect software package (StatsDirect Ltd., Cambridge, UK) and selecting the option called proportion meta-analysis. First, a sensitivity analysis was performed to check that the values did not significantly change after eliminating each study from the analysis. Then, publication bias was assessed using the Egger linear regression. The I2 index was used as a measure of heterogeneity. 

## 3. Results

The search returned a total of 398 articles, which, after eliminating duplicates, yielded 278 articles. After applying the inclusion and exclusion criteria, a total of 36 studies were obtained for full-text reading, and *n* = 16 studies were finally selected, of which 3 contributed data for the meta-analysis. The data from the study selection process are shown in Figure 1.

### 3.1. Characteristics of the Studies Included

A total of 7537 subjects were included in the sample, most of which were men. One study was quasi-experimental, and the other studies (*n* = 15) were cohort studies. Regarding the countries where the studies were performed, *n* = 3 were conducted in the USA (*n* = 3) [14,15,16] and *n* = 5 in Sweden [17,18,19,20,21]. Of the selected studies, which were performed between 1997 and 2020, 11 involved elective surgery [22,23,24,25]. The 16 studies evaluated quality of life prior to surgery, coinciding with the preoperative angiography performed at the surgical hospitalisation appointment. The follow-up time for most studies was six months [26,27,28]. The included studies used the following scales to measure quality of life: NHP (*n* = 8) and SF36 (*n* = 8). All the selected studies passed the critical reading for bias analysis. The characteristics of the studies and their main results are presented in Table 1.

### 3.2. Quality of Life Before and After a CABG

Studying cardiac, non-cardiac, preoperative, and early postoperative factors helps us know the health status of patients and predict their quality of life after surgery [22,26]. The quality of life of patients undergoing cardiac catheterisation improved dramatically between 6 weeks [23] and 3 months [17,26] or 6 months [15,21] after surgery, particularly with regard to the group of items encompassing physical problems [23]. Sexual health problems in men persisted throughout the follow-up period [17,18]. Physical problems improved according to the functional capacity of the patients prior to surgery [14]. Female sex [18,29], age, hypertension, obesity, renal failure, cerebrovascular disease, unstable angina [28], being a smoker, and having a psychiatric pathology [16] are all factors that have been shown to delay the recovery of post-surgery quality of life [19,20,24]. 

### 3.3. Differences in Quality of Life Scores Before and After a CABG

The studies that analysed quality of life using the SF questionnaire all showed quality improvements in both physical and mental aspects [14,15,16,22,23,24,25,26]. The least physical improvement was 2.2 points [14] and the most was 8.2 points [15,22]. For the mental aspect of quality of life, the improvement in the score ranged from a maximum of 3.6 points [22] to a minimum of 0.3 points [23]. Studies using the NHP questionnaire all showed improvements in quality of life scores with differences of up to 6 points after 10 years [19], 10 points after two years [17], and, in some cases, the differences in quality of life being greater for women than men [18].

### 3.4. Meta-Analysis for Estimating the Prevalence of Pre- and Post-CABG Impact on Quality of Life

Of the studies included in the systematic review, three contained the data necessary to perform the meta-analysis and used the second part of the NHP questionnaire. The total sample for the meta-analysis was *n* = 1997 people who received a CABG. With regard to the impact on the different aspects of quality of life analysed in part two of the NHP questionnaire (working life, work/housework, social life, family relationships, sex life, hobbies, and holidays), there was a decrease in the prevalence of impact on the seven areas before and after CBAG (Table 2). The I^2^ of the meta-analyses performed was over 90%. The Annex shows the Forestplots of impact prevalence of the seven areas before and after CABG (Figure 2, Figure 3, Figure 4, Figure 5, Figure 6, Figure 7 and Figure 8).

## 4. Discussion

The aim of the study was to analyse how coronary artery bypass grafting (CABG) influences quality of life. It has been observed that after CABG, in most studies, people exhibit significant improvements in the different dimensions of quality of life as analysed in the SF and NHP questionnaires. This positive result was also confirmed by the meta-analytical estimates of the impact on quality of life, with a lower prevalence of impact in all the dimensions of quality of life analysed. Within the SF, the physical dimension seems to improve more than the mental aspect.

A CABG seems to be very beneficial for patients, since in addition to the positive quality of life results, other studies indicate that it positively influences the occurrence of depression [30], can lead to the disappearance of symptoms for around 15 years [31], decreases death resulting from other causes, reduces hospital admission, and reduces death due to cardiovascular factors [32]. In addition, mortality in this type of surgery appears to be declining substantially [32]. Therefore, although surgery still involves risk and the possibility of future complications for individuals, it appears that the benefits are positive and appropriate in relation to the risk. These risks and complications seem to be reduced when the surgery is not performed urgently and when the patient presents no other pathologies [31].

The effects of CABG on more physiological aspects, such as the left ventricular ejection fraction, have also been analysed in other studies, which report improvements in those patients in whom the fraction was diminished before surgery, but a deterioration in those in whom the fraction was at normal levels [33]. Some authors also recommend performing a coronary angiography after the CABG to avoid the appearance of possible postoperative complications, as between 2% and 8% of heart attacks are reported in the perioperative period [34].

On the other hand, a significant difference between Percutaneous Coronary Intervention (PCI) and CABG regarding cardiovascular death has not been observed [35].

From a clinical perspective, this cardiac surgery, one of the most widely performed in the world, has a good scientific basis that supports the improvements it generates in quality of life and other aspects. For this reason, this type of surgery continues to be performed daily across the globe, and improvements are being researched with the use of existing technology in order to determine the optimal way to operate in the future, in the least invasive manner, and with the most lasting effects [4].

The main limitation of this study is the variability in the location and countries where the analysed studies were carried out. Therefore, depending on the country where the results are to be analysed or implemented, this factor should be taken into account. Additionally, there was only a limited number of studies with the necessary data to perform the meta-analysis, and the influence of other important factors on quality of life has not been included in many of these studies. Future research on the subject should include values correlating quality of life with other variables (like gender or age as indicated in some studies [18,28,29]) that allow the meta-analytical estimation of factors that may influence quality of life post-CABG, as well as experimental comparisons of how different CABG techniques or treatments influence quality of life, in order to determine the most cost-effective method.

## 5. Conclusions

The scientific literature shows that coronary artery bypass grafting improves a patient’s quality of life of in both the physical and mental aspects, although this improvement is more extensive with respect to physical factors. This favours normalisation of the day-to-day lives of these individuals in their personal and working environments, with a decreased prevalence of impact on the various aspects of life of between 18% and 6%. This operation seems to be a good choice for improving the quality of life of people with coronary pathologies once the possible existing risks have been assessed. 

## Figures and Tables

**Figure 1 ijerph-17-08439-f001:**
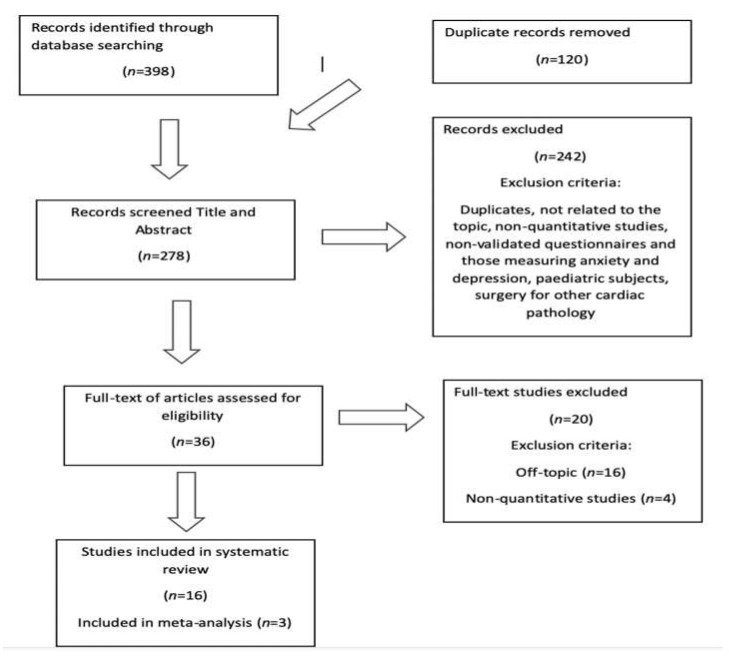
Study-selection diagram.

**Figure 2 ijerph-17-08439-f002:**
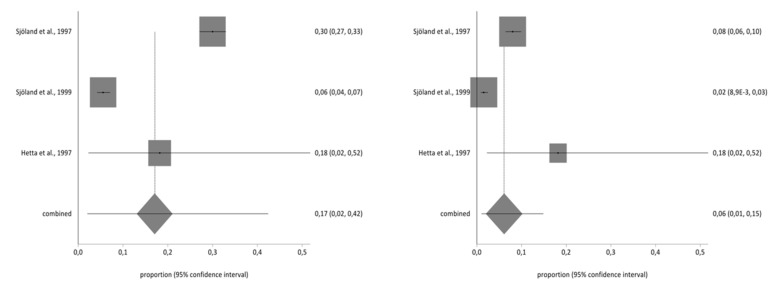
Pre- and post-impact on working life.

**Figure 3 ijerph-17-08439-f003:**
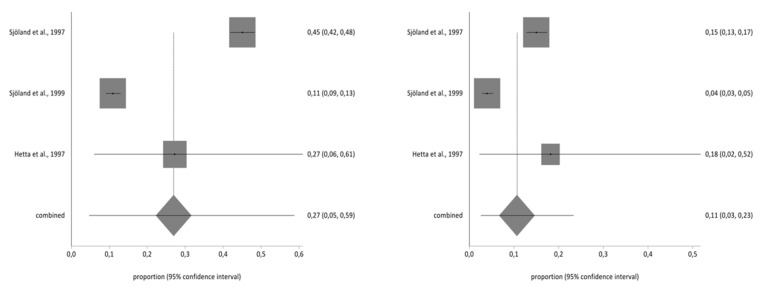
Pre- and post-impact on work/housework.

**Figure 4 ijerph-17-08439-f004:**
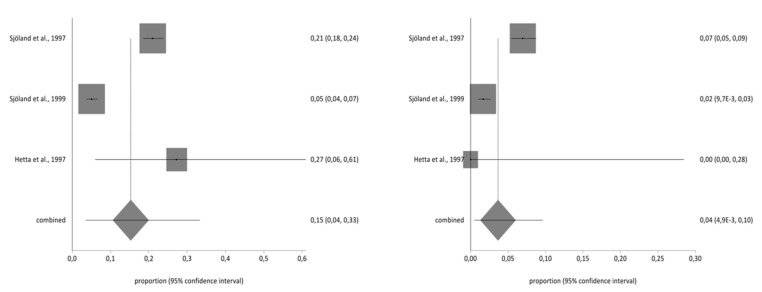
Pre- and post-impact on social life.

**Figure 5 ijerph-17-08439-f005:**
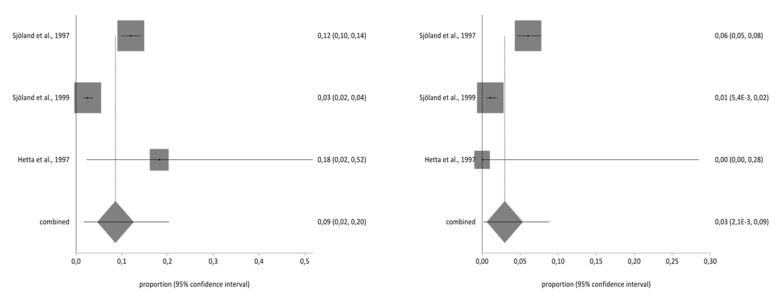
Pre- and Post-impact on family relationships.

**Figure 6 ijerph-17-08439-f006:**
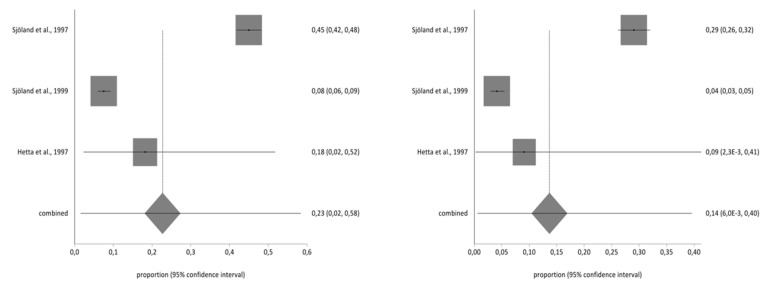
Pre- and post-impact on sex life.

**Figure 7 ijerph-17-08439-f007:**
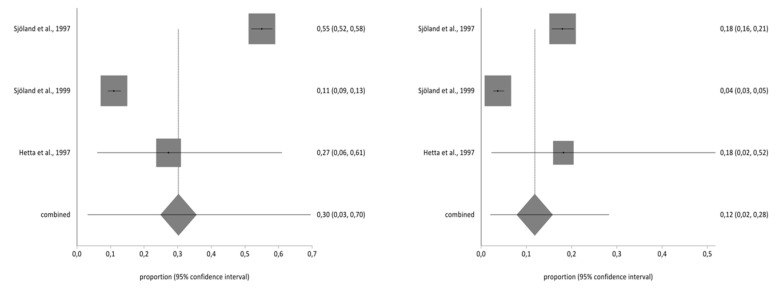
Pre- and post-impact on hobbies.

**Figure 8 ijerph-17-08439-f008:**
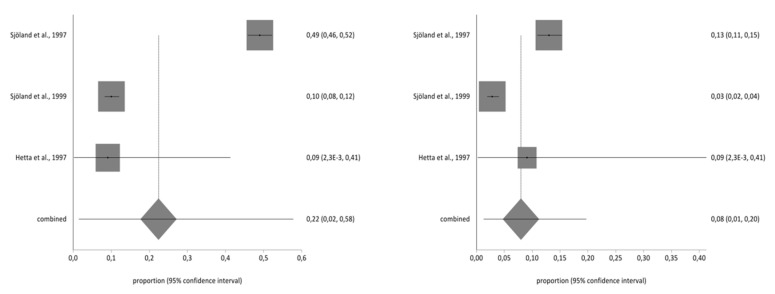
Pre- and post-impact on holidays.

**Table 1 ijerph-17-08439-t001:** Characteristics of the included studies (*n* = 16).

Study	Design	Sample	TYPE OF CABG	Quality of Life Measurement Questionnaire	Follow-Up	Average (DE) Pre	Average (DE) Post	Main Results	EL/RG
Lie et al [22], 2010 Norway	Prospective Cohorts	185 90% men	Elective	SF36	Beforehand, after 6 months	MSC 47.7 (11.2) PCS 39.0 (SD 10.2)	MSC 51.3 (10.7) PCS 47.2 (SD 9.8)	Studying cardiac, non-cardiac, preoperative and early postoperative factors helps us predict the quality of life of patients after surgery.	2b/B
Sjöland et al. [18], 1997 Sweden	Prospective Cohorts	Pre: 1160 Post-3 months: 1059, 1 year: 1045, 2 years: 1027 83% men	Emergency and Elective	NHP	Beforehand (at angiography appointment), after 3 months, 1 year, 2 years	20.5	3 months: 11.4 1 year: 11.9 2 years: 10.4	The greatest improvement in quality of life was at 3 months, for physical capacity and patient pain. Sexual problems persisted for 2 years after the surgery.	2b/B
Sandau et al. [14], 2007 USA	Prospective Cohorts	64 78.1% men	Elective	SF12 (short form of SF36)	72 h beforehand, after 3 months	MCS 49.6 (9.6) PCS 40.0 (10.6)	MCS 53.2 (9.5) PCS 42.2 (10.3)	Participants gained an average of 2.2 points (PCS) and 3.6 points (MCS). Although these changes appear small, the clinical significance of changes in an individual’s score depends largely on the functional capacity associated with the score.	2b/B
Ballan and Lee [23], 2007 Australia	Quasi-experimental	62 87.1% men	Elective	SF36	Beforehand, after 6 weeks	MSC 53.4 (12.7) PSC 26.1 (8.0)	MSC 53.7 (10.1) PSC 33.5 (10.2)	The PCS scores improved and were statistically significant 6 weeks after surgery. No significant differences were found in MCS scores.	1B/A
Herlitz et al. [19], 2003 Sweden	Prospective Cohorts	1225 (beforehand), 1358 (5 years), 976 (10 years) 98.5% men	Emergency and elective	NHP	Beforehand (during angiography), after 5 years, and 10 years	20.8	12.1 (5 years) 14.5 (10 years)	Patient quality of life improved, generally, at 10 years, despite increasing age. The scores for the second and third measurements deteriorated.	2b/B
Oreel et al. [24], 2020 The Netherlands	Prospective Cohorts	48 87.5% men	Elective	SF36	Beforehand, after 6 months	MCS 46.2(-) PCS 36(-)	MCS 51.9(-) PCS 43(-)	Quality of life was lower in women, and their physical health improved more slowly than that of male patients.	2b/B
Herlitz et al. [20], 2005 Sweden	Prospective Cohorts	637 75% men	(1) normal waiting list, (2) admitted patients, (3) patients with unstable angina, (4) emergency patients with unstable angiography, (5) emergency patients with acute myocardial infaction, (6) emergency patients with ventricular fibrillation	NHP	Beforehand, after 10 years	-	-	Being female, age, hypertension, obesity, renal failure, and cerebrovascular disease all play a role in the post-surgery recovery of quality of life.	2b/B
Neto et al. [25], 2010 Poland	Prospective Cohorts	44 59% men	Elective	SF36	Beforehand, after 3 and 6 months	-	-	The older population presents both cardiovascular and quality of life improvement after surgery. There are no statistically significant changes in the physical abilities of patients.	2b/B
Edell-Gustafsson et al [21], 1997 Sweden	Prospective cohorts (Pilot study)	6 beforehand 5 after 100% men	Elective	NHP	Two days beforehand, 1 month after	8.3	5.8	After a month, quality of life improved, although wound pain persisted influencing sleep quality.	2b/B
Grady et al. [15], 2011 USA	Prospective cohorts	136 70% men	Elective	SF36	Beforehand, after 3, 6, 12 months. Annually	MSC 51.88 (2.24) PSC 43.33 (2.73)	MSC 54.94 (1.61) PSC 51.65 (1.93)	There was an improvement in the quality of life between 3 and 6 months. After 3 years, it remained stable.	2b/B
Sjöland et al. [18], 1999 Sweden	Prospective cohorts	1160 83% men	-	NHP	Beforehand, 3 months, after 1 year, and 2 years	Men 19 Women 28	Men 10.4-8.7 Women 13.9-13.6	The women presented increased concomitant illnesses and a lower quality of life. The men encountered greater sexual problems prior to and 2 years after the surgery.	2b/B
Rumsfeld et al. [16], 2004 USA	Prospective cohorts	1973 99% men	-	SF36	Beforehand and after 6 months	MCS 44.3 PCS 33.0	MSC 46.1 PCS 38.2	Being a smoker and presenting a psychiatric pathology influences post-surgery quality of life.	2b/B
Mathisen et al. [26], 2007 Norway	Prospective cohorts	108 81% men	-	SF36 (General-care subscale)	Beforehand, after 3 months, 6 months, and 1 year	57.7 (21.1)	67.2 (19.7)	Quality of life can both influence and be used as a health status outcome after surgery. Most of the improvements in quality of life occurred in the first 3 months.	2b/B
Peric et al. [28], 2006 Serbia	Prospective cohorts	243 80% men	-	NHP	Beforehand and after 6 months	-	-	Patients with a higher degree of angina had worse quality of life both before and after the operation	2b/B
Peric et al. [27], 2005 Serbia	Prospective cohorts	243 80% men	Elective	NHP	Beforehand and after 6 months	-	-	Patients with a high mortality risk according to EUROSCORE have a worse quality of life before surgery and improved perceived energy after surgery.	2b/B
Peric et al. [29], 2010 Serbia	Prospective cohorts	243 80% men	Elective	NHP	Beforehand and after 6 months	-	-	Although the quality of life of both sexes improves after CABG, women have a worse quality of life both before and after surgery.	2b/B

Note: CABG = coronary artery bypass graftin; MSC = mental component of quality of life; NHP = Nottingham Health Profile; PCS = physical component of quality of life; SD = Standard deviation; SF = Short Form Health Survey.

**Table 2 ijerph-17-08439-t002:** Meta-analytical estimate of the impact on quality of life according to the 7 aspects of Part 2 of the NHP (*n* = 1997).

Dimension	Prevalence Pre (CI-95%)	Prevalence Post (CI-95%)
Impact on working life	17% (2–42%)	6% (1–15%)
Impact on work/housework	27% (5–59%)	11% (3–23%)
Impact on social life	15% (4–33%)	4% (1–10%)
Impact on family relationships	9% (2–20%)	3% (1–9%)
Impact on sex life	23% (2–58%)	14% (1–40%)
Impact on hobbies	30% (3–70%)	12% (2–28%)
Impact on holidays	22% (2–58%)	8% (1–20%)

Note. CI, confidence interval.

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
