# Peer review of "Quality of Life After Coronary Artery Bypass Surgery: A Systematic Review and Meta-Analysis"

_ijerph, 2020, doi:10.3390/ijerph17228439_

Round 1
Reviewer 1 Report
The manuscript "Quality of Life After Coronary Artery Bypass Surgery: A Systematic Review and Meta-Analysis" is a well-written manuscript and well describes various aspects of the quality of life after CABG surgery. But some modifications required to improve the manuscript quality.
In the manuscript, the abstract section should specify the time period of sampling selection.
In the abstract section, describe the major targets of the quality of life, altered significantly after CABG.
In methods section 2.5, describe briefly how you performed the statistical analysis and provide more details of the statistical package used in the study.
Did you notice any difference between sociodemographic factors and quality of life in these CABG patients after surgery?
The result section "3.3 Differences in quality of life scores before and after a CABG" should provide a supporting figure to differentiate the quality of life between women and men subjects.
Author Response
Dear reviewer,
Thank you for the review and for your comments. We think that our manuscript has improved with your suggestions. Please find below the response to each comment highlighted in yellow. All the changes in the manuscript have been tracked.
Kind regards.
The manuscript "Quality of Life After Coronary Artery Bypass Surgery: A Systematic Review and Meta-Analysis" is a well-written manuscript and well describes various aspects of the quality of life after CABG surgery. But some modifications required to improve the manuscript quality.
In the manuscript, the abstract section should specify the time period of sampling selection.
The information has been included in the abstract.
In the abstract section, describe the major targets of the quality of life, altered significantly after CABG.
The information has been included in the abstract.
In methods section 2.5, describe briefly how you performed the statistical analysis and provide more details of the statistical package used in the study.
More information has been included in the methodology.
Did you notice any difference between sociodemographic factors and quality of life in these CABG patients after surgery?
It was not an aim of the study and just a few studies included information about that. However the information about sociodemograhic factors included in the studies is shown in line 170, being female seems to be worse for quality of life recovery after surgery.
The result section “3.3 Differences in quality of life scores before and after a CABG” should provide a supporting figure to differentiate the quality of life between women and men subjects.
We cannot create a Figure about that information because we cannot meta-analyze those differences and because just a few studies include those differences.
Reviewer 2 Report
Interesting paper.
Some issues.
It should be better clarified the design of the study. Probably I would define as review rather than meta-analysis.
Comparison between CABG and PCI often lack data about quality of life. Quote on a recent meta-analysis (PMID: 32392283) and comment.
Author Response
Dear reviewer,
Thank you for the review and for your comments. We think that our manuscript has improved with your suggestions. Please find below the response to each comment highlighted in yellow. All the changes in the manuscript have been tracked.
Kind regards.
Interesting paper.
Some issues.
It should be better clarified the design of the study. Probably I would define as review rather than meta-analysis.
The study design is a systematic review with meta-analysis (the forestplot is a exclusive figure of the meta-analysis). Thus, we cannot delete the term meta-analysis.
Comparison between CABG and PCI often lack data about quality of life. Quote on a recent meta-analysis (PMID: 32392283) and comment.
This meta-analysis information has been included the discussion
Reviewer 3 Report
The paper concerns an interesting current problem related to the quality of life patients after CABG, widely used procedure which nowadays patients undergo both for diagnostic and therapeutic purposes.Therefore it is very important to assess the immediate and long-term effects of this procedure. One of the health and well-being indicator, that can be used in this evaluation is quality of life. It is a subjective indicator, but important from the patient's point of view. Authors, in their manuscript, tried to answer the question "how coronary artery bypass graft (CABG) influences quality of life". For this purpose they performed a systematic review and a meta-analysis according to the PRISMA recommendations.
I think that Authors missed to indicate the period of the papers included in the analysis as well as number of studies in particular kind: cohort studies, cross-sectional studies etc.In my opinion Authors should include also keyword: "follow-up study" which would expand the scope of the search (for example Perrotti A, Ecarnot F, Monaco F, Dorigo E, Monteleone P, Besch G, Chocron S. Quality of life 10 years after cardiac surgery in adults: a long-term follow-up study. Health Qual Life Outcomes. 2019 May 22;17(1):88. doi: 10.1186/s12955-019-1160-7).
I don't understand what Authors meant when they wrote: No restrictions were placed on the publication date, sample size, or type of surgery, whether elective or emergency, thus minimising publication bias I agree with the first part of this sentence but the second part? How it could minimise bias? It should be explained.
I have also question if Authors included any confounding factors (for example age, gender, clinical condition?). Whether information about these factors was available in the original papers? It should be disscussed. This lack is the weakness of this article, because quality of life after CABG could be influenced by a lot of factors.
I didn't find also in this manuscript an analysis of bias what is which is usually used in meta-analysis
Despite some doubts I rate this paper as interesting, because it is only a few similar papers but it needs improvement.
Author Response
Dear reviewer,
Thank you for the review and for your comments. We think that our manuscript has improved with your suggestions. Please find below the response to each comment highlighted in yellow. All the changes in the manuscript have been tracked.
Kind regards.
The paper concerns an interesting current problem related to the quality of life patients after CABG, widely used procedure which nowadays patients undergo both for diagnostic and therapeutic purposes.Therefore it is very important to assess the immediate and long-term effects of this procedure. One of the health and well-being indicator, that can be used in this evaluation is quality of life. It is a subjective indicator, but important from the patient's point of view. Authors, in their manuscript, tried to answer the question "how coronary artery bypass graft (CABG) influences quality of life". For this purpose they performed a systematic review and a meta-analysis according to the PRISMA recommendations.
I think that Authors missed to indicate the period of the papers included in the analysis as well as number of studies in particular kind: cohort studies, cross-sectional studies etc.In my opinion Authors should include also keyword: "follow-up study" which would expand the scope of the search (for example Perrotti A, Ecarnot F, Monaco F, Dorigo E, Monteleone P, Besch G, Chocron S. Quality of life 10 years after cardiac surgery in adults: a long-term follow-up study. Health Qual Life Outcomes. 2019 May 22;17(1):88. doi: 10.1186/s12955-019-1160-7).
We have included the period, the number of every kind of study and “follow-up study” to the keywords.
I don't understand what Authors meant when they wrote: No restrictions were placed on the publication date, sample size, or type of surgery, whether elective or emergency, thus minimising publication bias I agree with the first part of this sentence but the second part? How it could minimise bias? It should be explained.
Your suggestion is correct. We have reformulated the sentence.
I have also question if Authors included any confounding factors (for example age, gender, clinical condition?). Whether information about these factors was available in the original papers? It should be disscussed. This lack is the weakness of this article, because quality of life after CABG could be influenced by a lot of factors.
The influence of different factors in quality of life is not included in all studies and the information is shown in lines 169-173. Following your comments we have included information about in the discussion and also as a limitation of the study.
I didn't find also in this manuscript an analysis of bias what is which is usually used in meta-analysis
The analysis of bias is explained in methods point 2.4 and more information about it has been included in the results.
Despite some doubts I rate this paper as interesting, because it is only a few similar papers but it needs improvement.
Thank you for your comments and for your suggestions.
Round 2
Reviewer 1 Report
The manuscript "Quality of Life After Coronary Artery Bypass Surgery: A Systematic Review and Meta-Analysis" now looks interesting. The authors well answered most of the reviewers' comments and explained their limitations.